# USP26 suppresses type I interferon signaling by targeting TRAF3 for deubiquitination

**Cheng-Lan Sheng, Bang-Dong Jiang, Chun-Qiu Zhang, Jin-Hua Huang, Zi Wang, Chao Xu** *

Department of Clinical Laboratory, Chongming Brach Shanghai Tenth People's Hospital, Tongji University School of Medicine, Shanghai, P. R. China

* 18706190100@163.com

**Data Availability Statement:** All relevant data are within the paper.

**Funding:** Financial support was provided by the Science Foundation of Shanghai Health Commission, China (20194Y0108).

## Abstract

Deubiquitinating enzymes (DUBs) play a pivotal role in regulating the antiviral immune response by targeting members of the RLR signaling pathway. As a pivotal member of the RLR pathway, TRAF3 is essential for activating the MAVS/TBK-1/IRF3 signaling pathway in response to viral infection. Despite its importance, the function of DUBs in the TRAF3-mediated antiviral response is poorly understood. Ubiquitin-specific protease 26 (USP26) regulates the RLR signaling pathway to modulate the antiviral immune response. The results demonstrate that EV71 infection upregulates the expression of USP26. Knockdown of USP26 significantly enhances EV71-induced expression of IFN-β and downstream interferon-stimulated genes (ISGs). Deficiency of USP26 not only inhibits EV71 replication but also weakens the host's resistance to EV71 infection. USP26 physically interacts with TRAF3 and reduces the K63-linked polyubiquitination of TRAF3, thereby promoting pIRF3-mediated antiviral signaling. USP26 physically interacts with TRAF3 and reduces the K63-linked polyubiquitination of TRAF3, thereby promoting pIRF3-mediated antiviral signaling. Conversely, knockdown of USP26 leads to an increase in the K63-linked polyubiquitination of TRAF3. These findings unequivocally establish the essential role of USP26 in RLR signaling and significantly contribute to the understanding of deubiquitination-mediated regulation of innate antiviral responses.

## 1 Introduction

EV71 infection has been consistently associated with neurological disorders since the first outbreak was documented by American researcher Schmidt in California in 1974 [1]. Subsequent outbreaks of EV71 infection have been reported by numerous countries following the isolation of the virus from patients. Currently, China publishes annual reports of such outbreaks [2]. The treatment and prevention of EV71 infection rely on a range of effective measures, including antibiotics, vaccines, drugs that synthesize viral RNA, and exogenous cytokines [3]. It is imperative that further research be conducted into EV71 infection and its pathophysiology in order to enhance the effectiveness of treatments and reduce mortality rates. The current efficacy of treatments remains suboptimal.

**Competing interests:** The authors have declared that no competing interests exist.

The innate immune system serves as the primary defense against invading viruses by identifying pathogen-associated molecular patterns (PAMPs) [4]. The human body contains a multitude of pattern recognition receptors (PRRs),including retinoic acid-inducible gene I (RIG-I)-like receptors (RLRs), Toll-like receptors (TLRs), NOD-like receptors (NLRs), C-type lectin receptors (CLRs), and cytoplasmic DNA recognition receptors [5]. The activation of these receptor proteins initiates the appropriate natural immune response, which serves to resist infection by pathogenic microorganisms. RLRs, one of the four major pattern recognition receptors, comprise three members: RIG-I, MDA5, and LGP2 [6–8]. RIG-I/MDA5 recognize RNA viruses that interact with the mitochondrial antiviral signaling (MAVS) adapter molecule, which in turn recruits complexes such as tumor necrosis factor receptor-associated factor 3 (TRAF3) and TRAF2/6 [9]. Subsequently, the TBK1-mediated IRF3 axis or the IKKs-mediated NF-κB axis is employed, resulting in the production of IFN-I, pro-inflammatory cytokines, and chemokines in response to viral infection [10].

Ubiquitination is a vital post-translational modification that preserves cellular homeostasis during various biological processes, including the immune response [11], cell division, growth, and apoptosis [12–14]. K48-linked polyubiquitin chains regulate the proteasomal degradation of target proteins at lysine residues [15]. Monoubiquitin or polyubiquitin chains serve to attach other ubiquitins. K63-linked and linear polyubiquitin chains regulate protein kinase activation and cell signaling [16]. The process of ubiquitination is highly reversible, as deubiquitinases (DUBs) are capable of removing ubiquitin chains. Based on their structural properties, there are six groups of DUBs, with over one hundred known DUBs. The largest family of deubiquitinases (DUBs) is the ubiquitin-specific proteases (USPs), which play a crucial role in antiviral immune signaling pathways, particularly during EV71 infection. For example, USP19 specifically inhibits IRF3 activation induced by EV71 by removing the K63 ubiquitin chain on TRAF3 [17]. Moreover, USP24 limits the K63 ubiquitin chain on TBK1, thereby promoting EV71 infection [18]. In contrast, USP4 cleaves the K48 ubiquitin chains on TRAF6, thereby promoting the RIG-I-mediated IFN-I antiviral immune response [19]. Deubiquitination is a precisely regulated process that plays a pivotal role in the antiviral response, a mechanism employed by the host.

Ubiquitin-specific protease 26 (USP26) is a prominent member of the USP family of deubiquitinating enzymes (DUBs). USP26 was initially discovered by Wang, who successfully isolated this gene from mouse spermatogonia [20]. USP26 plays a crucial role in regulating the occurrence and development of tumors [21–24], bone homeostasis [25] and male infertility [26–28]. The present study demonstrates that EV71 upregulates the expression of USP26 in human rhabdomyosarcoma cells (RD), suggesting a potential role for USP26 in regulating viral infections. This is because we have identified the mechanism by which the reduction of USP26 leads to an increase in the production of IFN-I, which inhibits EV71 infection. The targeting of USP26, an unidentified regulator utilized by EV71 to evade host antiviral defenses, represents a potential therapeutic avenue for the treatment of EV71 infection.

## 2 Materials and methods

### 2.1. Cell culture

HEK293T cells and the RD cells (National Collection of Authenticated Cell Cultures) were cultured in Dulbecco modified Eagle medium (DMEM) (HyClone, USA) replenished with 10% fetal bovine serum (Gibco, USA) and 100 U/ml Penicillin–Streptomycin Solution (Invitrogen, USA) at 37˚C under a 5% $CO_2$ atmosphere.

## 2.2. Virus infection

The EV71 strain (BrCr strain, ATCC VR784, GeneBank accession number: U22521) was provided by China Center for Type Culture collection. For viral infection, RD cells were infected with EV71 at the multiplicity of infection (MOI) value of 1.0 in half-volume 2% serum-reduced medium to allow for viral adsorption for 1.5 h, followed by replacement with full-volume maintenance medium for different durations. Infection of EV71 for 0 h in the experiments represents that cells were mock-infected with equal volume of phosphate-buffered saline (PBS). And the viral protein VP1 expression was used as a control for each experiment involving the viral infection.

## 2.3. Plasmids and reagents

The pcDNA3.1-Flag-RIG-I, pcDNA3.1-Flag-MDA5, pcDNA3.1-Flag-MAVS, pcDNA3.1-Flag-TBK1, pcDNA3.1-Flag-TRAF3, pcDNA3.1-Flag-TRAF6, and pcDNA3.1-Flag-IRF3 were purchased from Transheep Bio (Shanghai, China). All the plasmids were confirmed by sequencing. The transient transfection was carried out by Lipofectamine 3000 (Invitrogen, CA, USA) in accordance with standard protocols.

Antibodies against IRF3 (4302), phosphorylated IRF3 (4947), TRAF6 (8028), TRAF3 (3504) and horseradish peroxidase (HRP) conjugated goat anti-rabbit IgG secondary antibodies(L3012) were purchased from the Cell Signaling Technology (CST, USA). Antibodies for USP26 (13126-1-AP), anti-GAPDH and anti-hemagglutinin (HA)-HRP (561–7) were obtained from the Protein TECH Group (Chicago, USA) and MBL (Japan) respectively; EV71/VP1 (169442), K63 linkage-specific ubiquitin (179434) and horseradish peroxidase (HRP) conjugated goat anti-mouse IgG secondary antibodies (97023)were from Abcam (UK); antibodies ubiquitin (SC-8017) was from Santa Cruz Biotechnology (USA); anti-Flag (M2)-horseradish peroxidase (A8592) was from Sigma (USA); anti-rabbit-IgG (sc-52336) was from Santa Cruz (USA). Poly(I:C) of high molecular weight (HMW) was purchased from Invivogen (CA, USA). Transient gene silencing with small-interfering RNA (SiRNA) was performed using INTERFERin (Polyplus Transfection, Illkirch, France). The sequence for silencing USP26 (Gene ID: 83844) was synthesized by RiboBio (Guangzhou, China), and listed as follows:

SiRNA1:5′-GCUCGCAGAUGUGUAACCU-3′;
SiRNA2:5′-AAACAGAUCUGGUUCACUU-3′;
SiRNA3:5′-GCACAAGACUUCCGUUGGA -3′;
Scrambled control sequences (SiNC): 5′-UUCUCCGA ACGMGUCA CGU-3′.

## 2.4. EV71 plaque assay

RD cells were infected with EV71 (MOI = 1.0). After 24 h, the supernatant was collected in EP tubes. RD cells in a 96-well plate were infected with an equal dilution of viral supernatant for 1.5 h. Next, the viral supernatant was discarded, and cells were washed with $1 \times$ PBS, and maintained in DMEM with 10% fetal bovine serum (FBS). The cell morphology was observed and recorded every day. The $TCID_{50}$ was calculated using the Spearman-Karber algorithm.

## 2.5. RNA quantitation

The total RNA was extracted by the Trizol method and reversely transcribed using a Prime Script RT Reagent kit (TaKaRa, Japan). Quantitative real-time PCR (RT-qPCR) was performed using an ABI 7500 PCR system (Applied Biosystems, USA). The GAPDH gene was used to make a control. Fluorescent dye for amplification was used for SYBR Green Premix Ex Taq II

PCR mix (TaKaRa, Japan). The reaction conditions were: pre-denaturation at 95˚C for 10 mins, denaturation at 95˚C for 10 s, annealing at 60˚C for 1 min, and extension at 72˚C for 30 s, 40 cycles. The CT value at the end of the reaction was used to analyze the expression levels of the target genes. The calculation method of the target gene is as follows: three replicate wells are set for each sample, and the relative expression level of the target gene is calculated by the $2^{-\Delta\Delta Ct}$ method ($\Delta\Delta Ct$ = (Ct target gene–Ct reference gene) experiment- (Ct target gene–Ct internal reference gene) control). RT-qPCR primers are listed in Table 1.

## 2.6. Immunoblot, co-immunoprecipitation and ubiquitination assays

The experiments were performed as previously described [19]. Equal amounts of cell lysates were resolved using 8±15% polyacrylamide gels transferred to PVDF membrane. Membranes were blocked in 5% non-fat dry milk in PBST and incubated overnight with the respective primary antibodies at 4˚C. The membranes were incubated at room temperature for 1 h with appropriate HRP-conjugated secondary antibodies and visualized with Plus-ECL according to the manufacturer's protocol. For immunoprecipitation assays, the lysates were immunoprecipitated with IgG or the appropriate antibodies and protein G Sepharose beads. The precipitates were washed three times with lysis buffer containing 500 mM NaCl, followed by immunoblot analysis. For deubiquitination assays, the cells were lysed with the lysis buffer and the supernatants were denatured at 95 ˚C for 5 min in the presence of 1% SDS. The denatured lysates were diluted with lysis buffer to reduce the concentration of SDS below 0.1% followed by immunoprecipitation with the indicated antibodies. The immunoprecipitates were subjected to immunoblot analysis with anti-ubiquitin chains.

**Table 1. Primer pairs for real-time PCR.**

| Gene | Sequence |
|---|---|
| USP26(H) | F: CCTCTCCATCAACCTTCCCCAA<br>R: CTCCAACGGAAGTCTTGTGCTC |
| USP26(M) | F: GAGGAAGAGCATAGACCCAGTG<br>R: TGGACGGCTTTGAGTAAGTGCC |
| EV71/VP1 | F: GAGTGGCAGATGTGATTGA<br>R: TCCAGTGTCTAAGCGATGA |
| IFN-β(H) | F: CCCTATGGAGATGACGGAGA<br>R: CTGTCTGCTGGTGGAGTTCA |
| IFN-β(M) | F: GCCTTTGCCATCCAAGAGATGC<br>R: ACACTGTCTGCTGGTGGAGTTC |
| MX1(H) | F: GGCTGTTTACCAGACTCCGACA<br>R: CACAAAGCCTGGCAGCTCTCTA |
| MX1(M) | F: TGGACATTGCTACCACAGAGGC<br>R: TTGCCTTCAGCACCTCTGTCCA |
| ISG15(H) | F: CTCTGAGCATCCTGGTGAGGAA<br>R: AAGGTCAGCCAGAACAGGTCGT |
| ISG15(M) | F: CATCCTGGTGAGGAACGAAAGG<br>R: CTCAGCCAGAACTGGTCTTCGT |
| Viperin(H) | F: CCAGTGCAACTACAAATGCGGC<br>R: CGGTCTTGAAGAAATGGCTCTCC |
| Viperin(M) | F: GGAAGGTTTTCCAGTGCCTCCT<br>R: GGAAGGTTTTCCAGTGCCTCCT |
| GAPDH (H) | F: TATGACAACAGCCTCAAGA<br>R: ATGAGTCCTTCCACGATAC |
| GAPDH (M) | F: CATCACTGCCACCCAGAAGACTG<br>R: ATGCCAGTGAGCTTCCCGTTCAG |

## 2.7. Transfection and luciferase reporter assay

RD cells were co-transfected with the IFN-β promoter, ISRE, IRF3 promoter and NF-κB promoter luciferase reporter plasmid and a TK-Renilla luciferase reporter, together with vector alone SiUSP26 constructs. 24 h later, cells were infected with EV71 for 12 h. Luciferase reporter activities were measured in triplicate using the Dual-Luciferase reporter assay system (Promega, Madison, WI, USA), according to the manufacturer's protocol, and quantified using the 96-well plate luminometer (Promega). The firefly luciferase to Renilla luciferase ratios were determined and were defined as the relative luciferase activity.

## 2.8. Mice

C57BL/6 mice (Wild type, WT) were from the Lab Animal Center of Soochow University (Suzhou, China). The Usp26$^{-/-}$ mice on a C57BL/6J background were generated by Cyagen Biosciences Inc. (Guangzhou, China) using CRISPR—Pro technology. The experimental animals were housed and bred in a barrier facility at the Medical Laboratory Animal Center of Shanghai Tenth People's Hospital of Tongji University in a 12-h light/12-h dark cycle in accordance with the Code of Ethical Management of Laboratory Animals of Shanghai Tenth People's Hospital of Tongji University. Mice used in the experiment were euthanized by head and neck dislocation after intraperitoneal injection of 1.25% tribromoethanol anesthetic. The mice used in the experiments lived in a specific pathogen-free (SPF) grade barrier environment. The temperature was maintained at 20–26 degrees Celsius. In accordance with national standards, the mice were fed commercial laboratory animal feed that was sterilized by cobalt 60 irradiation. The mouse drinking water was pH 2.5–3.0 acidified. All surgeries were performed after anesthesia with 1.25% tribromoethanol anesthetic, and every effort was made to minimize pain in the mice.

## 2.9. CCK8 assay

RD cells were first transfected with different constructs. After centrifuging at 800 ×$g$ for 3 mins at 25°C, the number of cells was calculated using a counting plate. RD cells were seeded at a proportion of $5 \times 10^3$/well in 96-well plates. The media was changed to DMEM with 2% FBS. One group was infected with EV71 (MOI = 1.0) at different time points, and another group was infected by EV71 at different MOI. Meanwhile, the control group (containing only cells) and blank holes (containing only culture medium) were set up. After 24 h, CCK-8 solution (10 μL) was immediately added to each well. The plate was put in an incubator at 37°C and incubated for 1 h. Absorbance at 450 nm was measured with a microplate reader. The results were calculated by the following formula: cell survival = [(Absorbance of test group−Absorbance of blank) / (Absorbance of control group−Absorbance of blank)] × 100%.

## 2.10. Enzyme-linked immunosorbent assay (ELISA)

Cytokine production in supernatants of in vitro cell cultures or sera of mice was measured by ELISA of mouse IFN-β (ExCell Bio, China) according to the manufacturer's protocol.

## 2.11. Statistical analysis

The data were presented as means ± standard deviation (SD). Differences were evaluated using one-way ANOVA or t-test, implemented in SPSS Statistics 22.0 software, with a threshold of $P < 0.05$ for determining inferential statistical significance.

## 3 Results

### 3.1. USP26 expression is significantly increased during EV71 infection

PCR microarray analysis revealed that EV71 infection downregulated the expression of host deubiquitinases, including USP54, USP43, and USP4, while upregulating the expression of USP26, USP5, USP24, USP19, and TNFAIP3 was observed to varying degrees (Fig 1A). The study provides clear evidence that USP26 plays a regulatory role in the innate immune response against EV71 infection. The upregulation of USP26 expression during EV71 infection, as evidenced by RT-qPCR (Fig 1B) and Western blot (Fig 1C), corroborates the significance of USP26 in this process. These findings provide compelling evidence that EV71 infection induces the expression of USP26 in RD cells.

### 3.2. USP26 depletion inhibits EV71 infection

In order to investigate the regulatory role of USP26 in EV71 replication, three SiRNAs were designed with the objective of reducing USP26 expression. the most effective SiRNA for knocking down USP26 expression was identified as SiRNA (#2) (Fig 2A and 2B). We used SiUSP26 (#2) to observe the effects of USP26 on EV71 infection and antiviral immune signaling. The results of this study demonstrate that the knockdown of USP26 during EV71 infection results in a time-dependent decrease in the expression of EV71-encoded VP1 proteins (Fig 2C). USP26 knockdown consistently downregulated EV71 viral RNA levels (Fig 2D) and viral titers (Fig 2E), thereby demonstrating its potential to enhance host cell resistance to EV71 infection. Further experiments confirmed that USP26 knockdown alleviated disease

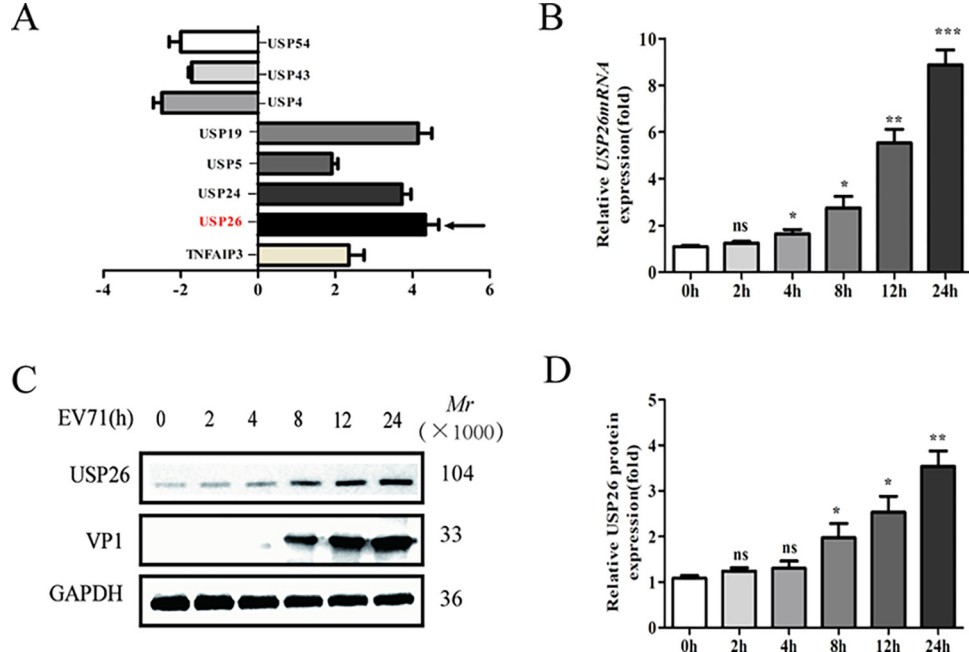

**Fig 1. USP26 expression is significantly increased during EV71 infection.** (A) Differential gene expression of human DUBs in RD cells infected with EV71 after 8 h were analyzed by PCR microarray. (B) RT-qPCR analysis of the USP26 mRNA levels in RD cells infected by EV71 (MOI = 1.0) at different time points. (C) Western blot analysis of the expression of USP26 and EV71/VP1 in RD cells infected by EV71 (MOI = 1.0) at different time points. The independent experiments were performed in triplicate. (D) Data were shown relative to GAPDH expression and are presented as the mean ± SD from three independent experiments. NS, not significant $P>0.05$, $*P<0.05$, $** P<0.01$ and $*** P<0.001$.

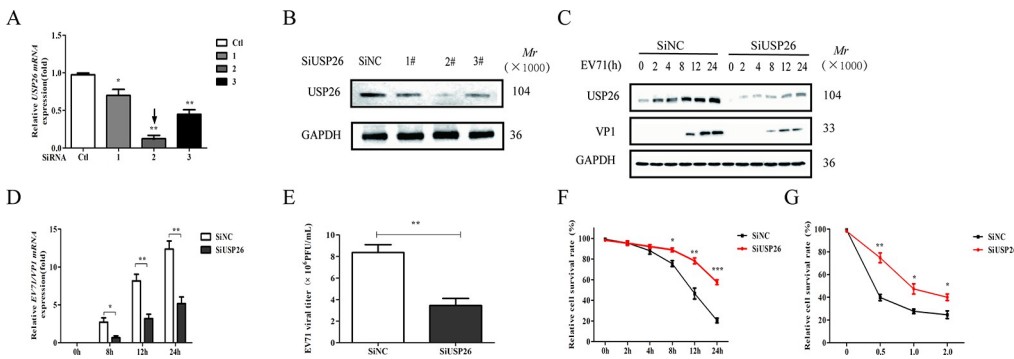

**Fig 2. USP26 depletion inhibits EV71 infection.** (A, B) Western blot (A) and RT-qPCR (B) analysis of USP26 levels in RD cells transfected with control SiRNA (SiNC) or USP26-specific SiRNAs (SiUSP26, #1, #2 and #3) for 48 h. (C) Western blot analysis of EV71 protein levels in RD cells transfected with SiNC or SiUSP26(#2) and then infected by EV71 (MOI = 1.0) for different times. (D) RT-qPCR analysis of EV71-VP1 RNA levels in RD cells transfected with SiNC or SiUSP26 (#2) and then infected by EV71 (MOI = 1.0) for indicated times. (E) Viral titers of EV71 in RD cells with stable knockdown of USP26 were analyzed. (F) CCK8 analysis of RD cell survival after infection with EV71 (MOI = 1.0) for the indicated times. The cell survival rates at 0 h time points were normalized as 100%. (G) CCK8 analysis of RD cells survival after infection with different amount of EV71. The cell survival rates when MOI = 0 was normalized as 100%. Data were shown as the mean ± SD from three independent experiments. NS, not significant $P > 0.05$ * $P < 0.05$, ** $P < 0.01$ and *** $P < 0.001$. The independent experiments were performed in triplicate.

progression caused by EV71 infection in RD cells, promoting cell survival (Fig 2F and 2G). These findings provide compelling evidence that USP26 exerts a negative regulatory effect on EV71 infection.

## 3.3. Knocking down USP26 results in the upregulation of type I IFN production

EV71 is known to trigger cellular antiviral IFN-I signaling through MDA5. Moreover, HMW Poly(I:C) has been demonstrated to activate type I IFN signaling via MDA5 [29]. To further assess the physiological function of UPS26, EV71 infection and transfection of poly(I:C) resulted in higher ISER, IFN-β and IRF3 reporter activities in knockdown of USP26 (Fig 3A–3C). However, no difference was observed in NF-κB reporter activities in the knockdown of USP26 (Fig 3D). USP26 knockdown was found to upregulate EV71-induced activation of ISRE, an important response element in triggering the expression of ISGs following the activation of IFN. It is crucial to highlight that this outcome was observed without any influence on the NF-κB signaling pathway. The data provides compelling evidence that USP26 significantly affects the downstream cascade that amplifies type I IFN expression (Fig 3E–3H). This indicates that the knockdown of USP26 significantly enhances the EV71-induced expression of ISGs (ISG15, MX1, and Viperin). To ascertain whether EV71 infection initiates type I IFN signaling, RD cells were infected with EV71 for varying time periods. As illustrated in Fig 3I, type I IFN signaling was indeed activated upon EV71 infection, while EV71 suppressed IRF3 activation during sustained infection. The study aimed to investigate the effects of USP26 knockdown on the activation of type I IFN signaling. The results showed that the knockdown of USP26 significantly increased EV71-induced phosphorylation levels of IRF3, while having no effect on protein levels of IRF3 (Fig 3J). These findings demonstrate the crucial role of USP26 in regulating the activation of type I IFN signaling during EV71 infection. Moreover, knockdown of USP26 was found to promote the mRNA expression of IFN-β induced by EV71 infection (Fig 3K), and the secreted IFN-β also increased in EV71-infected cells when USP26 was knocked down (Fig 3L). The data clearly demonstrate that knockdown of cellular USP26 significantly enhances the type I IFN signaling pathway.

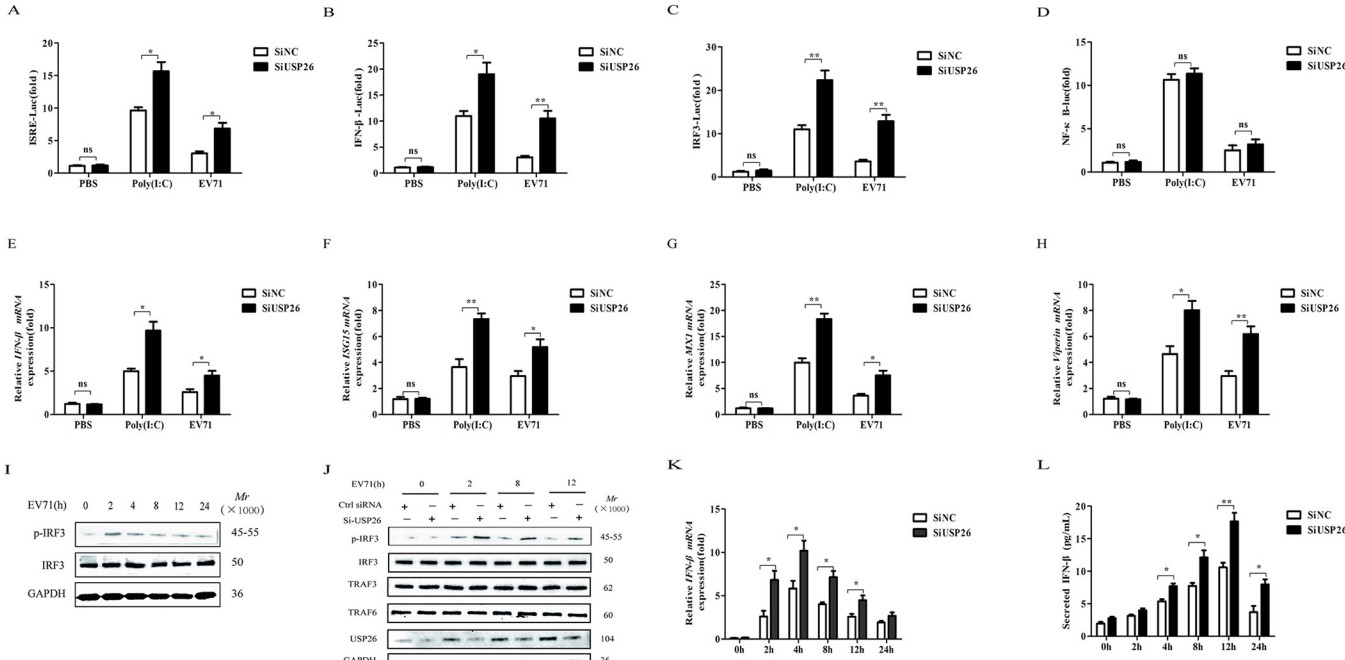

**Fig 3. Knocking down USP26 results in the upregulation of type I IFN production.** RD cells transfected with knockdown of USP26 or SiNC were infected with EV71(MOI = 1.0) or transfection with Poly (I:C) at 12h. (A-D) Luciferase assay was performed to analyze regulation of IFN-β, ISER, NF-κB and IRF3 promoter by knockdown of USP26. (E-H) The expression of IFN-β, ISG15, MX1 and Viperin mRNA levels were analyzed by RT-qPCR. (I) Western blot analysis of RD cells infected with EV71 for different duration of time, The cell lysates were analysed by immunoblotting using anti-pIRF3, anti-IRF3 and anti-GAPDH antibodies respectively. (J) Western blot analysis of p-IRF3 (pSer396-IRF3), total IRF3, TRAF3, TRAF6 levels in RD cells with knockdown of USP26 that were infected with EV71 (MOI = 1.0) at the indicated time. (K) RT-qPCR analysis of IFNβ mRNA levels in RD cells with knockdown of USP26 that infected with EV71 (MOI = 1.0) at the indicated time. (L) ELISA analysis of the IFN-β protein levels in RD cells with stable knockdown of USP26 that infected with EV71 (MOI = 1.0) at the indicated time. Data were shown relative to GAPDH expression and are presented as the mean ± SD from three independent experiments. NS, not significant $P>0.05$, ** $P<0.01$ and *** $P<0.001$. The independent experiments were performed in triplicate.

## 3.4. USP26-deficient mice were resistant to virus infection

To ascertain the role of USP26 in virus infection in vivo, we infected wild-type (WT) and USP26-/- mice were infected via tail vein injection with EV71 and their survival monitored. These findings provide strong evidence for the pivotal role of USP26 in the pathogenesis of EV71 infection. The results clearly demonstrate that USP26-/- mice were significantly more resistant to lethal EV71 infection than WT mice (Fig 4A–4C). Furthermore, interferon-stimulated gene (ISG) transcription was markedly increased in the serum of Usp26-/- mice infected with EV71 (Fig 4E–4G). The serum of USP26-/- mice with EV71 infection showed increased expressions of IFN-β (Fig 4D and 4H). These findings indicate that USP26 plays a pivotal role in host defense against RNA viruses in vivo, as evidenced by its ability to negatively regulate the expression of downstream genes induced by the virus.

## 3.5. USP26 can interact with TRAF3

The RLR signaling pathway plays a pivotal role in the IFN-I antiviral immune response against EV71 invasion. The present study examines the impact of USP26 on the RLR signaling pathway during EV71 infection. Our findings provide valuable insights into the role of USP26 in the RLR signaling pathway during EV71 infection. HEK293T cells were transfected with the key components of the RIG-I/MAVS signaling pathway, including RIG-I, MDA5, MAVS, TRAF3, TRAF6, TBK1 and IRF3. Subsequently, immunoprecipitation (IP) experiments were conducted to identify the proteins that interact with USP26 (Fig 5A–5G). The study provides

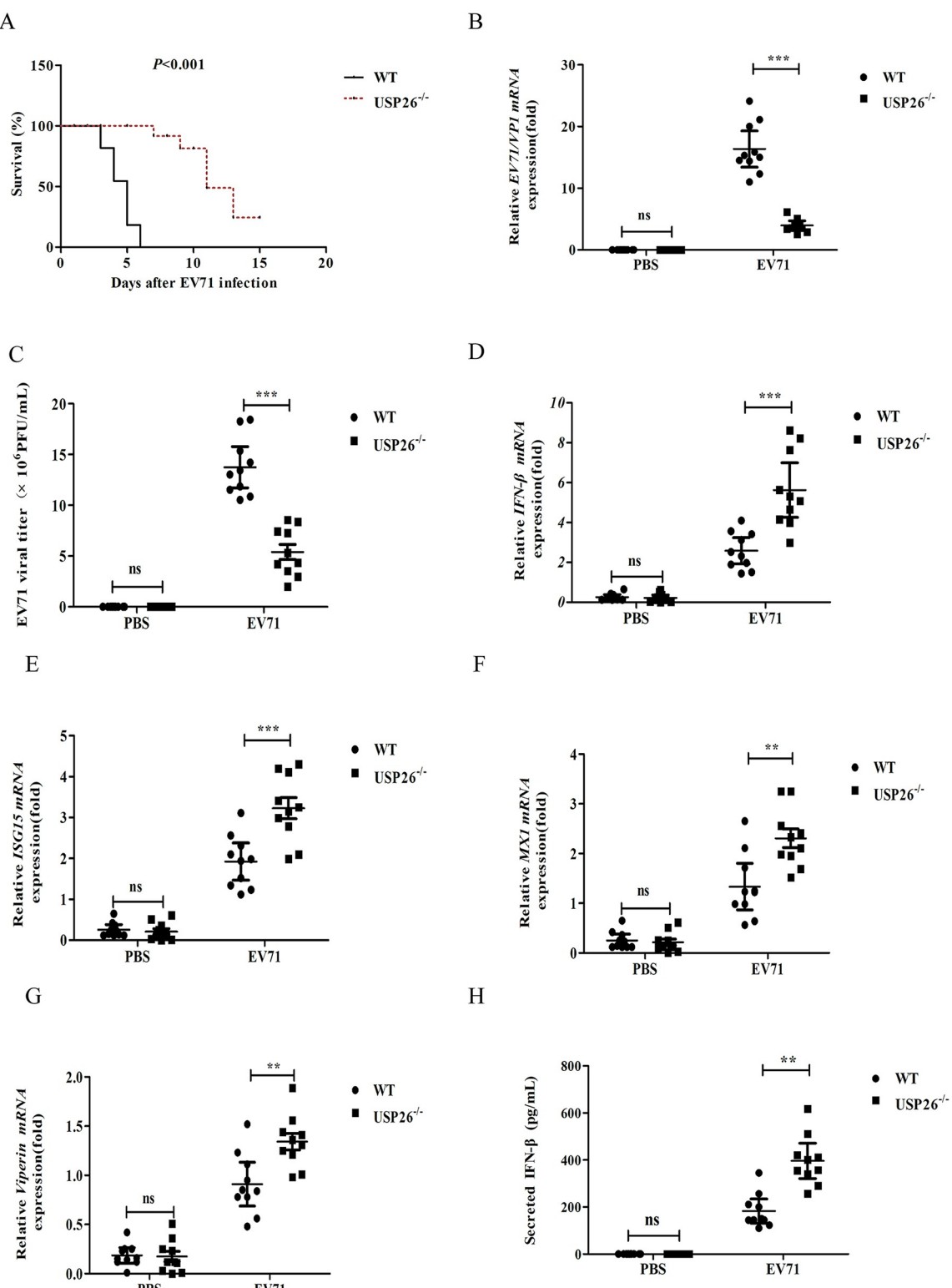

**Fig 4. USP26-deficient mice were resistant to virus infection.** WT and USP26$^{-/-}$ mice were infected with EV71 virus ($1 \times 10^7$ PFU per mouse). (A) Survival rate of WT and USP26$^{-/-}$ mice after EV71 infection for indicated time. (B) Viral EV71 VP1 mRNA was determined by RT-qPCR. (C) Viral titres of EV71 virus were determined. (E) Expression of IFN-β and IL-6 was determined by qPCR. (D-G) Expression of IFN-β, ISG15, MX1 and Viperin were determined by RT-qPCR. (H) Serum levels of IFN-β was determined by ELISA at 3 days post infection. Data shown are the mean ±SD. NS, not significant $P > 0.05$, ** $P < 0.01$ and *** $P < 0.001$. Data are representative of three independent experiments with similar results.

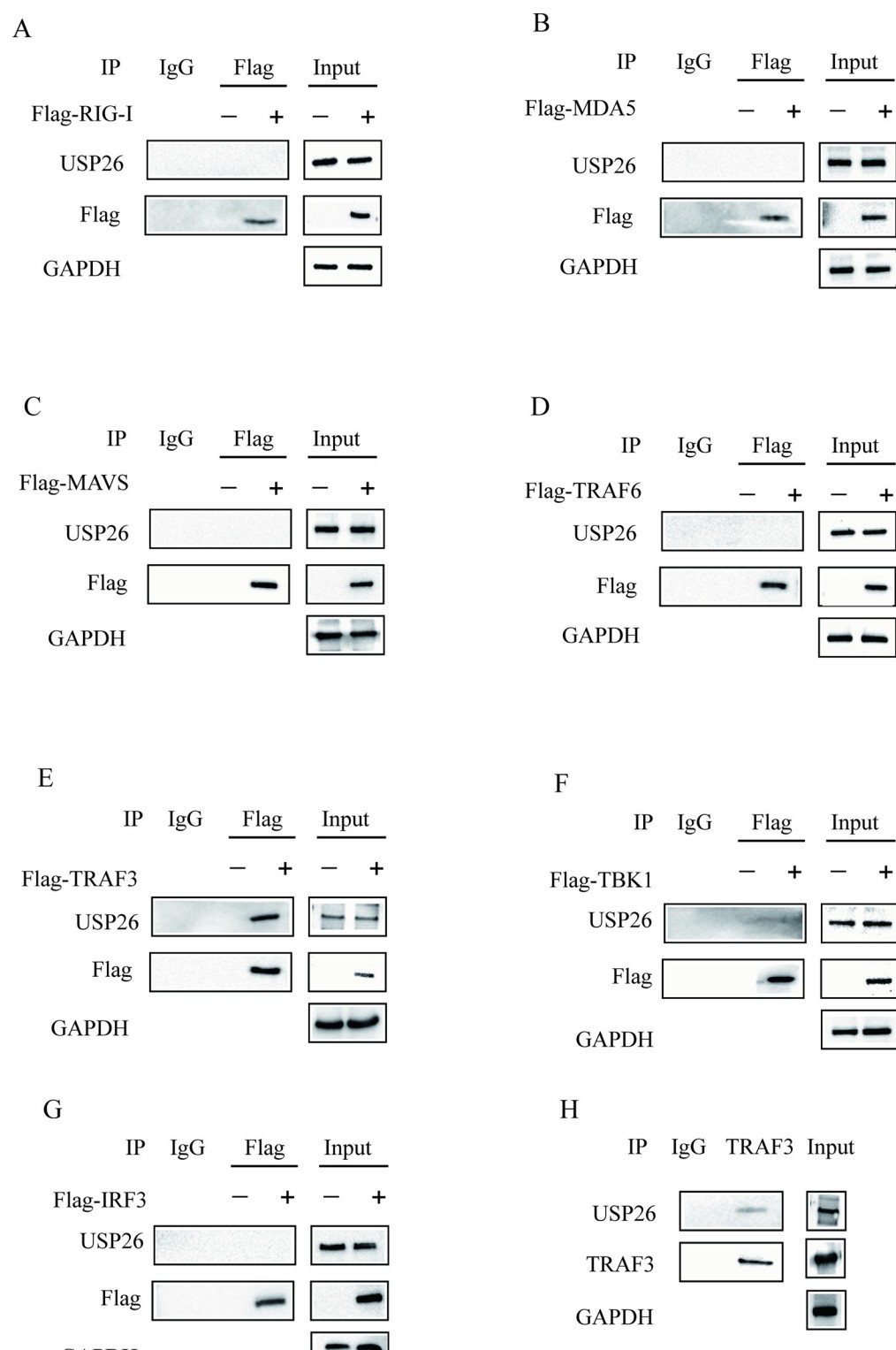

**Fig 5. USP26 can interact with TRAF3.** (A-G) Co-immunoprecipitation analysis of the interaction between USP26 and the key components of RIG-I/MAVS signaling. HEK293T cells were transfected with Flag-RIG-I or Flag-MDA5 or Flag-MAVS or Flag-TRAF6 or Flag-TRAF3 or Flag-TBK1 or Flag-IRF3, and immunoprecipitated with anti-Flag agarose beads. The eluted immunocomplex was then subjected to SDS-PAGE analysis with anti-USP26 antibody. (H) Co-immunoprecipitation analysis the interaction between endogenous USP26 and TRAF3 in HEK293T cells. The independent experiments were performed in triplicate.

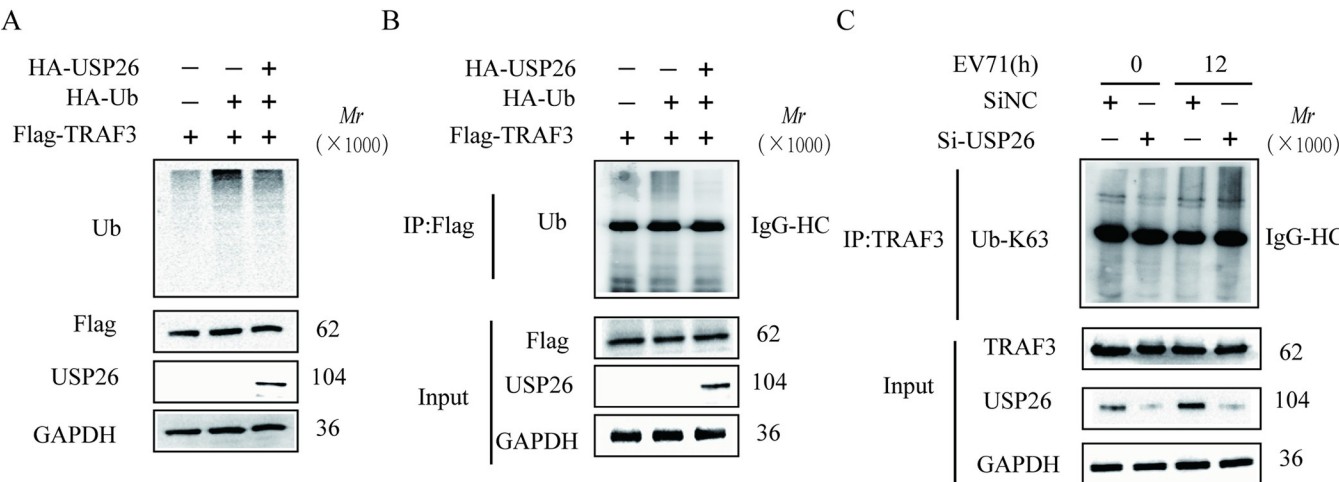

**Fig 6. USP26 decreases the polyubiquitination of TRAF3.** (A) Western blot analysis of HEK293T cells transfected with Flag-TRAF3, HA-USP26 and HA-Ub for cellular ubiquitination levels; GAPDH was used as a loading control. (B) HEK293T cells transfected with Flag-TRAF3, HA-Ub, together with wild-type HA-USP26 were collected and immunoprecipitated with anti-Flag agarose beads. The eluted immunocomplex was then subjected to SDS-PAGE analysis with antiubiquitin antibody. (C) Western blot analysis of RD cells transfected with knockdown of USP26 or SiNC for 48 h, followed by treatment with EV71 (MOI = 1) for 12 h. The cell lysates were subjected to immunoprecipitation with anti-TRAF3 antibody, followed by western blot analysis of eluted immunocomplex with K63 linkage-specific ubiquitin antibodies.

clear evidence that USP26 interacts with TRAF3, but not with RIG-I, MDA5, MAVS, TRAF6, TBK1, or IRF3. Moreover, the study provides evidence that endogenous TRAF3 interacts with endogenous USP26 in HEK293T cells (as depicted in Fig 5H). These findings provide compelling evidence that USP26 plays a pivotal role in regulating IRF3 activation and IFN-I production by targeting TRAF3.

### 3.6. USP26 decreases the polyubiquitination of TRAF3

The studies conducted thus far have demonstrated that USP26 interacts with TRAF3. Consequently, the effect of USP26 on TRAF3 was investigated. As a deubiquitinase, USP26 significantly reduces the overall ubiquitination levels of the cells, as demonstrated in Fig 6A. To confirm the deubiquitinating effect of USP26 on TRAF3, a pull-down assay was conducted on Flag-tagged TRAF3 molecules, and the impact of USP26 on TRAF3's deubiquitination was observed. Furthermore, the levels of endogenous ubiquitination of TRAF3 following USP26 knockdown were assessed (Fig 6B). The results in Fig 6C provide compelling evidence that USP26 can regulate the K63-linked polyubiquitination of TRAF3, which in turn affects IRF3 activation and IFN-I production during EV71 infection. Specifically, the K63-linked polyubiquitination levels of TRAF3 were significantly elevated in EV71-stimulated USP26-knockdown cells (lane 4).

## 4 Discussion

The RIG-I and MDA5 proteins are capable of recognizing viral RNA through the CTD domain, with each protein recognizing a distinct set of setviruses. EV71 is a positive-sense, single-stranded RNA virus that belongs to the Picornavirus genus of enteroviruses [30]. Based on current knowledge, it is believed that EV71 can only be recognized by MDA5. Kuo transfected HeLa and RD cells with EV71 RNA, thereby inducing MDA5 activation and subsequently activating IRF3 and IFN-β transcription. This unequivocally confirms that MDA5 is the crucial molecule that recognizes EV71 and efficiently activates downstream antiviral signals [31].

Nevertheless, EV71 has developed a highly effective immune evasion strategy. The research involved screening 88 deubiquitinating enzymes in EV71-infected RD cells. It was found that the expression of USP26 significantly increased at 8 hours post-infection. As the infection progressed, the expression of VP1 in EV71 exhibited a gradual increase. USP26 plays a pivotal role in the antiviral immune response. EV71 is renowned for its sophisticated and potent immune escape mechanisms, which inhibit the antiviral immune response of host cells. Consequently, it is of the utmost importance to identify novel targets that can enhance antiviral immune activity.

TRAF3 plays a pivotal role in virus-induced type I IFN signaling as an indispensable adaptor molecule for RLR-mediated IRF3 activation [32]. Studies have demonstrated that virus-triggered K63-linked polyubiquitination of TRAF3 by cIAP-1 and cIAP-2 is necessary for the induction of cellular antiviral responses [33]. OTUB1 [34], USP19 [17], and USP25 [35] are deubiquitinases that regulate cellular antiviral innate immunity by modulating TRAF3. It is of paramount importance to emphasize that TRAF3 is meticulously regulated during viral infections, a phenomenon that is not unexpected given its pivotal role in this process. The study demonstrates that USP26 interacts with TRAF3 and reduces its K63-linked polyubiquitination. The knockdown of USP26 has been observed to increase TRAF3 K63-linked polyubiquitination and enhance pIRF3-mediated antiviral signaling.

The production of Type I interferon is triggered by viral infections, which represents a crucial mechanism for antiviral innate immunity and late-stage adaptive immunity [36]. However, the excessive production of IFNs or proinflammatory cytokines can have destructive effects on the host. Consequently, a successful immune response against viral infections must be tightly regulated [37]. The Lys63-linked polyubiquitination of TRAF3 represents a pivotal event in the RLR-mediated antiviral response [17]. The study demonstrated that USP26 targets the TRAF3 protein and affects its K63-linked polyubiquitination in cells infected with EV71. The upregulation of USP26 during EV71 infection suggests the existence of a novel regulatory mechanism within the RIG-I/MAVS signaling pathway. The USP26-mediated regulation of TRAF3 ubiquitination and IRF3 activation results in a weakening of the host's antiviral innate immunity, which ultimately benefits the immune escape of EV71. To substantiate the correlation between USP26 and EV71 infection, USP26 was transfected into EV71-infected RD cells. The knockdown of USP26 resulted in a reduction in the mRNA and protein expression of VP1 in EV71, which inhibited EV71 replication and reduced RD cell apoptosis. These findings provide compelling evidence that USP26 plays a pivotal role in enhancing the ability of RD cells to defend against EV71. It is plausible that USP26 plays a role in regulating the infection of other viruses. This could impact both host and virus-encoded proteins, ultimately promoting or inhibiting infection through various mechanisms. Future research must investigate the role of USP26 in diverse viral infections.

A correlation has been demonstrated between the genetic diversity of patients and their susceptibility to EV71 infection, as well as the severity of the disease. For instance, patients with congenital TLR3 or MDA5 defects were found to be more susceptible to EV71 infection [38]; Genetic polymorphisms in the endothelial nitric oxide synthase gene were also found to correlate with the degree of EV71 infection in Chinese children [39]. It is evident that there is no correlation between children with USP26 defects or mutations and EV71 infection. The study provides compelling evidence demonstrates that USP26 plays a role in the production of IFN-I during EV71 infection, offering protection against EV71 infection in vivo and vitro. It is therefore postulated that individuals with USP26 deficiency or mutation may exhibit resistance to EV71 infection. Inhibiting the deubiquitinase activity of USP26 could markedly enhance the antiviral response and confer benefits patients with compromised immune systems. Further

clinical cases and large-scale epidemiological screening studies are required to substantiate this hypothesis.

## 5 Conclusion

In conclusion, the results of the study demonstrate that USP26 has the capacity to suppress RLR-mediated innate antiviral signaling. The removal of USP26 has been demonstrated to activate the RLR-MAVS-IRF3 pathway initiated by RNA viruses and to enhance the expression of downstream genes. Moreover, the absence of USP26 results in a reduction in EV71 replication and an attenuation of the host's susceptibility to EV71 infection. In terms of its mechanistic action, USP26 functions as a negative regulator of the cellular type I IFN antiviral immune response to EV71 infection, specifically by targeting TRAF3. USP26 is essential for maintaining the delicate balance of the virus-induced type I IFN signaling pathway. Further research is necessary to fully comprehend the multifaceted and intricate significance of this multifunctional protein.

## Supporting information

**S1 Raw images.**
(PDF)

## Acknowledgments

We thank Doctor Yang for his advices on EV71 propagation and technical assistance.

## Author Contributions

**Conceptualization:** Chao Xu.

**Data curation:** Chao Xu.

**Formal analysis:** Cheng-Lan Sheng, Bang-Dong Jiang, Zi Wang.

**Investigation:** Bang-Dong Jiang, Chao Xu.

**Methodology:** Cheng-Lan Sheng, Bang-Dong Jiang, Zi Wang.

**Resources:** Chun-Qiu Zhang.

**Software:** Cheng-Lan Sheng, Chun-Qiu Zhang, Zi Wang.

**Supervision:** Cheng-Lan Sheng, Zi Wang.

**Validation:** Chun-Qiu Zhang, Jin-Hua Huang.

**Visualization:** Jin-Hua Huang.

**Writing – original draft:** Chao Xu.

**Writing – review & editing:** Jin-Hua Huang, Chao Xu.

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
