## [Decision Letter · Decision Letter 0]

23 Apr 2024

PONE-D-24-12016USP26 suppresses type I interferon signaling by targeting TRAF3 for deubiquitinationPLOS ONE

Dear Dr. xu,

Thank you for submitting your manuscript to PLOS ONE. After careful consideration, we feel that it has merit but does not fully meet PLOS ONE’s publication criteria as it currently stands. Therefore, we invite you to submit a revised version of the manuscript that addresses the points raised during the review process.

**The current version of manuscript can be further improved by addressing the reviewer's comments. **

We look forward to receiving your revised manuscript.

Kind regards,

Kin-Hang Kok, PhD

Academic Editor

PLOS ONE

3. To comply with PLOS ONE submissions requirements, in your Methods section, please provide additional information regarding the experiments involving animals and ensure you have included details on (1) methods of sacrifice, (2) methods of anesthesia and/or analgesia, and (3) efforts to alleviate suffering.

“Financial support was provided by the Science Foundation of Shanghai Health Commission, China  (20194Y0108)”

“Financial support was provided by the Science Foundation of Shanghai Health Commission, China (20194Y0108), Chongming science and technology support plan (CKY2022-43). The authors have no other relevant affiliations or financial involvement with any organization or entity.”

“Financial support was provided by the Science Foundation of Shanghai Health Commission, China  (20194Y0108)”

7. PLOS ONE now requires that authors provide the original uncropped and unadjusted images underlying all blot or gel results reported in a submission’s figures or Supporting Information files. This policy and the journal’s other requirements for blot/gel reporting and figure preparation are described in detail at https://journals.plos.org/plosone/s/figures#loc-blot-and-gel-reporting-requirements and https://journals.plos.org/plosone/s/figures#loc-preparing-figures-from-image-files. When you submit your revised manuscript, please ensure that your figures adhere fully to these guidelines and provide the original underlying images for all blot or gel data reported in your submission. See the following link for instructions on providing the original image data: https://journals.plos.org/plosone/s/figures#loc-original-images-for-blots-and-gels.

8. We notice that your supplementary table is included in the manuscript file. Please remove them and upload them with the file type 'Supporting Information'. Please ensure that each Supporting Information file has a legend listed in the manuscript after the references list.

Additional Editor Comments:

The manuscript can be further improved by addressing the reviewer 1's comments.

Reviewers' comments:

Reviewer's Responses to Questions

**Comments to the Author**

1. Is the manuscript technically sound, and do the data support the conclusions?

Reviewer #1: Yes

2. Has the statistical analysis been performed appropriately and rigorously? 

Reviewer #1: Yes

3. Have the authors made all data underlying the findings in their manuscript fully available?

Reviewer #1: Yes

4. Is the manuscript presented in an intelligible fashion and written in standard English?

Reviewer #1: Yes

5. Review Comments to the Author

Reviewer #1: The current manuscript describes a study to USP26 which facilitates EV17 infection by antagonizing type I IFN through inactivating and deubiqtuinating TRAF3. Generally, the study is sound, and the evidence look solid. The use of siRNA knockdown to USP26 well address the proviral effect to EV70 in RD cells. The result of in vivo model study using USP26 knockout C57 mice is consistent to the in vitro study. Therefore, it clearly shows that USP26 is proviral to EV71. One question at this point is that why USP26 did not show up in the last publication of the same lab (PMID: 28391724). What was the coverage and replicability of the array study? An explanation and limitation can be provided. Mechanistically, it was suggested that USP26 interacted and deubiquitinated TRAF3. The data is presented logically. However, the role of TRAF3 to USP26 IFN antagonism is still not very clear. Quality of data should be improved. Overall, the work is solid with clear phenotype. Improvement can be made to the mechanistic studies. Some more concerns are listed below:

1. Only when some experiments are provided can link USP26 to RLR signaling and TRAF3. It is possible that Poly I:C and EV71 can activate non-RLR signaling in RD cells, such as toll-like receptor signaling (although should be minimal? Should provide evidences or references). In addition, no evidence was provided to show siUSP26 promotes type I IFN production through TRAF3. For example, would siTRAF3 but not siTRAF6 abolish the IFN induction by siUSP26?

2. Figure 5F is suboptimal. A band is merely observed in IP/USP26 blot.

3. Figure 5G, input immunoblots have error. Flag-IRF3 should be expressed in one control only? Instead, USP26 was only found in the IRF3 overexpressed cells. Shall the immunoblots be reversed? It will be better to provide size indication to each band of the IP experiment.

4. Figure 6A-6C are suboptimal. Where is the mono-Ub on figure 6A? Moreover, please provide size for all immunoblots. Poly-Ub smear should appear above the unlinked protein.

5. Line 247-249 is duplicated. The description to 3A-D was incomplete. Please revise.

6. Section 3.3, the use of the term ‘type I IFN signaling’ might be confusing. As the study focused on IRF3 activation, it is better to use ‘type I IFN production’ instead.

6. PLOS authors have the option to publish the peer review history of their article (what does this mean?). If published, this will include your full peer review and any attached files.

Reviewer #1: **Yes: **Cheung Pak Hin Hinson

---

## [Author Response · Author response to Decision Letter 0]

25 Jun 2024

We have revised the manuscript to meet the comments

---

## [Decision Letter · Decision Letter 1]

11 Jul 2024

USP26 suppresses type I interferon signaling by targeting TRAF3 for deubiquitination

PONE-D-24-12016R1

Dear Dr. xu,

We’re pleased to inform you that your manuscript has been judged scientifically suitable for publication and will be formally accepted for publication once it meets all outstanding technical requirements.

Kind regards,

Kin-Hang Kok, PhD

Academic Editor

PLOS ONE

Additional Editor Comments (optional):

The revised manuscript has been largely improved as suggested by reviewer comments. It is recommended for publication.

Reviewers' comments:

Reviewer's Responses to Questions

**Comments to the Author**

1. If the authors have adequately addressed your comments raised in a previous round of review and you feel that this manuscript is now acceptable for publication, you may indicate that here to bypass the “Comments to the Author” section, enter your conflict of interest statement in the “Confidential to Editor” section, and submit your "Accept" recommendation.

Reviewer #1: All comments have been addressed

2. Is the manuscript technically sound, and do the data support the conclusions?

Reviewer #1: Yes

3. Has the statistical analysis been performed appropriately and rigorously? 

Reviewer #1: Yes

4. Have the authors made all data underlying the findings in their manuscript fully available?

Reviewer #1: Yes

5. Is the manuscript presented in an intelligible fashion and written in standard English?

Reviewer #1: Yes

6. Review Comments to the Author

Reviewer #1: Main: No further comments.

1: The method of transfecting HMW poly I:C for activation of cytosolic MDA5 can be mentioned clearly. The method was absent.

2: Problem solved with the original film photos provided.

3: Problem solved in the revised manuscript.

4: Size was provided in the raw data.

5: Problem solved.

6: Problem solved.

7. PLOS authors have the option to publish the peer review history of their article (what does this mean?). If published, this will include your full peer review and any attached files.

Reviewer #1: No

---

## [Editor Report · Acceptance letter]

17 Jul 2024

PONE-D-24-12016R1 

PLOS ONE

Dear Dr. Xu, 

I'm pleased to inform you that your manuscript has been deemed suitable for publication in PLOS ONE. Congratulations! Your manuscript is now being handed over to our production team.

Kind regards, 

on behalf of

Dr. Kin-Hang Kok 

Academic Editor

PLOS ONE